# The Inhibiting Effect of GB-2, (+)-Catechin, Theaflavin, and Theaflavin 3-Gallate on Interaction between ACE2 and SARS-CoV-2 EG.5.1 and HV.1 Variants

**DOI:** 10.3390/ijms25179498

**Published:** 2024-08-31

**Authors:** Chung-Kuang Lu, Jrhau Lung, Li-Hsin Shu, Hung-Te Liu, Yu-Huei Wu, Yu-Shih Lin, Yao-Hsu Yang, Yu-Heng Wu, Ching-Yuan Wu

**Affiliations:** 1Department of Chinese Medicine, Chiayi Chang Gung Memorial Hospital, Chiayi 613, Taiwanr95841012@cgmh.org.tw (Y.-H.Y.); 2Department of Research and Development, Chiayi Chang Gung Memorial Hospital, Chiayi Branch, Putzu 613, Taiwan; jrhaulung@gmail.com; 3Department of Pharmacy, Chiayi Chang Gung Memorial Hospital, Chiayi 613, Taiwan; yohimba@cgmh.org.tw; 4School of Chinese medicine, College of Medicine, Chang Gung University, Taoyuan 333, Taiwan; 5Health Information and Epidemiology Laboratory, Chang Gung Memorial Hospital, Chiayi 613, Taiwan; 6Institute of Communications Engineering, The National Yang Ming Chiao Tung University, Hsinchu City 30010, Taiwan; 7Research Center for Chinese Herbal Medicine, College of Human Ecology, Chang Gung University of Science and Technology, Taoyuan 333, Taiwan

**Keywords:** SARS-CoV-2, spike protein, GB-2, omicron variants, theaflavin 3-gallate, theaflavin

## Abstract

The ongoing COVID-19 pandemic, caused by SARS-CoV-2, continues to pose significant global health challenges. The results demonstrated that GB-2 at 200 μg/mL effectively increased the population of 293T-ACE2 cells with low RBD binding for both SARS-CoV-2 Omicron EG.5.1 and HV.1 variants by dual-color flow cytometry, indicating its ability to inhibit virus attachment. Further investigation revealed that (+)-catechin at 25 and 50 μg/mL did not significantly alter the ACE2–RBD interaction for the EG.5.1 variant. In contrast, theaflavin showed inhibitory effects at both 25 and 50 μg/mL for EG.5.1, while only the higher concentration was effective for HV.1. Notably, theaflavin 3-gallate exhibited a potent inhibition of ACE2–RBD binding for both variants at both concentrations tested. Molecular docking studies provided insight into the binding mechanisms of theaflavin and theaflavin 3-gallate with the RBD of EG.5.1 and HV.1 variants. Both compounds showed favorable docking scores, with theaflavin 3-gallate demonstrating slightly lower scores (−8 kcal/mol) compared to theaflavin (−7 kcal/mol) for both variants. These results suggest stable interactions between the compounds and key residues in the RBD, potentially explaining their inhibitory effects on virus attachment. In conclusion, GB-2, theaflavin, and theaflavin 3-gallate demonstrate significant potential as inhibitors of the ACE2–RBD interaction in Omicron variants, highlighting their therapeutic promise against COVID-19. However, these findings are primarily based on computational and in vitro studies, necessitating further in vivo research and clinical trials to confirm their efficacy and safety in humans.

## 1. Introduction

As of May 2024, the global cumulative number of confirmed COVID-19 cases has exceeded 775 million, with more than 7 million deaths reported [1] by the World Health Organization (WHO). These figures highlight that SARS-CoV-2 continues to spread globally, and continued monitoring and preventive measures remain crucial. SARS-CoV-2, the virus responsible for COVID-19, is a member of the Coronaviridae family and belongs to the β-coronavirus genus. It is an enveloped virus with a positive-sense single-stranded RNA genome. SARS-CoV-2 has several key structural proteins, including the spike (S), envelope (E), membrane (M), and nucleocapsid (N) proteins. The spike protein is crucial for the virus’s ability to infect host cells. Its receptor-binding domain (RBD) directly interacts with the angiotensin-converting enzyme 2 (ACE2) receptor on the surface of human cells, facilitating viral entry [2,3].

SARS-CoV-2 variants arise from mutations in the virus’s genome, leading to changes in its characteristics such as transmissibility, virulence, and resistance to vaccines or treatments. The virus’s spike protein, which binds to the ACE2 receptor on host cells, is a key area where these mutations occur [4,5,6]. Some variants, like Omicron variants, have multiple mutations in RBD of the spike protein, which can decrease the effectiveness of antibodies and increase the virus’s infectivity and immune escape capabilities [4,5,6,7,8]. This raises concerns about vaccine efficacy and the need for booster doses or updated vaccines. The SARS-CoV-2 Omicron variant (B.1.1.529) was first identified in South Africa in November 2021 and quickly spread worldwide. It contains over 30 mutations in the spike protein, particularly in the receptor-binding domain, which enhance its ability to evade the immune system and increase its transmissibility. Omicron’s mechanism for cell entry has also changed, relying more on the endosomal pathway, allowing it to infect host cells more efficiently [9,10]. The SARS-CoV-2 Omicron EG.5.1 variant, also known as “Eris”, is a descendant of the XBB.1.9.2 subvariant. It features key mutations such as F456L and Q52H in the spike protein, which improve the virus’s ability to evade the immune system and bind more effectively to cell receptors, enhancing its transmissibility. These mutations, particularly F456L (part of the “Flip” mutations), have been associated with immune escape, enabling the variant to spread more effectively even among populations with prior immunity from vaccination or past infections. EG.5.1 was first reported in early 2023 and has rapidly spread across multiple countries. The World Health Organization has designated it as a “variant of interest” due to its increased prevalence and potential impact on public health. It has been linked to modest increases in COVID-19 cases and hospitalizations in regions such as Japan, New Zealand, South Korea, the UK, and the US [11]. The SARS-CoV-2 Omicron HV.1 variant, also known as EG.5.1.6.1, carries additional mutations that enhance its ability to enter human cells. Similar to EG.5.1, it has the F456L mutation and other modifications that help it escape immune detection. These mutations contribute to its higher infectivity and potential immune evasion, making it a variant of interest in ongoing genomic surveillance. The HV.1 variant emerged later in 2023 and quickly became a notable variant in the United States, among other regions. It has shown a rapid increase in prevalence second only to other major variants like EG.5.1. Its rise has prompted close monitoring and evaluation by health authorities to understand its potential impact on public health and vaccine efficacy [12,13,14]. The capacity of these variants to evade immune defenses creates obstacles in managing emerging outbreaks. This situation underscores the importance of continuous monitoring and potential adjustments to preventive measures.

Traditional Chinese medicines are widely used to treat various diseases in Taiwan [15,16]. Among these, GB-2 is a frequently used herbal formulation that contains *Glycyrrhiza uralensis* Fisch. and *Camellia sinensis* var. *assamica* (J.W. Mast.) Kitam. Previous studies have shown that GB-2 can reduce ACE2 protein levels in both cell and animal models without adverse effects [17]. However, the effects and mechanisms of GB-2 and its active components on the SARS-CoV-2 Omicron EG.5.1 and HV.1 variants have not been studied. (+)-Catechin, theaflavin, and theaflavin-3-gallate are key polyphenolic components predominantly present in tea with a particularly high concentration in *Camellia sinensis* var. *assamica* (J.W. Mast.) Kitam. These polyphenols are recognized for their antioxidant capabilities and associated health benefits. These compounds play a crucial role in the health-promoting properties of tea, particularly black tea, by potentially reducing oxidative stress, supporting cardiovascular health, offering antiviral effects, and providing protection against various diseases [18,19]. Moreover, glycyrrhizic acid is the major component of *Glycyrrhiza uralensis* Fisch. Therefore, this research investigated the inhibitory effects of GB-2, (+)-catechin, theaflavin, and theaflavin 3-gallate and glycyrrhizic acid on interaction between ACE2 and RBD of the SARS-CoV-2 Omicron EG.5.1 and HV.1 variants.

## 2. Results

### 2.1. Identification of Reference Compounds in GB-2 by HPLC Analysis

GB-2 primarily consists of four key components: (+)-catechin (Figure 1A), theaflavin 3-gallate (Figure 1B), theaflavin (Figure 1C), and glycyrrhizic acid (Figure 1D), which were used as reference standards. The presence of (+)-catechin, theaflavin 3-gallate, and theaflavin in GB-2 was confirmed through high-performance liquid chromatography (HPLC) analysis, as evidenced by the validated fingerprint chromatography (Figure 1E). Furthermore, glycyrrhizic acid, a compound found in *Glycyrrhiza uralensis* Fisch., was also verified as a reference compound in GB-2 using HPLC analysis (Figure 1F). Detailed information regarding the chromatography is shown in Appendix A.

### 2.2. Effect of GB-2 on the Interaction between ACE2 and RBD with the EG.5.1 and HV.1 Variants

The EG.5.1 and HV.1 variants are sublineages of the Omicron variant known for their high transmissibility [12,14,20]. GB-2 can inhibit the interaction between ACE2 and RBD with the mutations K417N–E484K–N501Y, K417N, N501Y, and L452R in a dose-dependent manner [21]. The effects of GB-2 on the interaction between ACE2 and the EG.5.1 and HV.1 variants were studied using dual-color flow cytometry [22]. Following treatment with various concentrations of GB-2, cell populations expressing MYC-tagged ACE2 or GFP-tagged RBD on their surface were examined, showing either high or low binding to the RBD with different variants, utilizing fluorescence-activated cell sorting. The low and high groups in our FACS analysis represent distinct cell populations based on their level of RBD binding. The “low” group is of particular interest, as it represents cells where the ACE2–RBD interaction has been inhibited or reduced by the test compounds. This inhibition is the primary effect we are investigating. The biological significance of the low-cell cluster is that it demonstrates the compounds’ ability to interfere with the virus’s attachment process. Cells in this cluster have reduced RBD binding, suggesting that these compounds are effectively blocking the interaction between the viral spike protein and the ACE2 receptor. This is crucial because preventing this interaction is a key strategy for inhibiting viral entry into host cells. In this study, the analysis indicated 200 μg/mL of GB-2 led to an increase in the population of 293T-ACE2 cells with low RBD binding for the Omicron EG.5.1 (Figure 2A,B) and HV.1 (Figure 2C,D) variants. These findings imply that GB-2 can block the interaction between ACE2 and the spike protein in the EG.5.1 and HV.1 variants.

### 2.3. Effect of (+)-Catechin and Glycyrrhizic Acid on the ACE2–RBD Interaction of Omicron Variants

Omicron EG.5.1 and HV.1 strains show reduced ACE2 interaction due to GB-2 effects (Figure 2). Consequently, we examined how (+)-catechin and glycyrrhizic acid, key components in GB-2, influence binding between these variants and ACE2. Our study employed dual-color flow cytometry for this analysis. The results revealed that (+)-catechin, tested at 25 and 50 μg/mL levels, did not modify the quantity of 293T-ACE2 cells displaying minimal RBD attachment for EG.5.1 (Figure 3A). Similarly, glycyrrhizic acid at identical concentrations failed to alter cell numbers, exhibiting low receptor-binding domain connections with both EG.5.1 and HV.1 types (Figure 3B,C).

### 2.4. Effect of Theaflavin on the ACE2–RBD Interaction of Omicron EG.5.1 and HV.1 Variants

Using dual-color flow cytometry, the inhibitory effects of theaflavin on the ACE2–RBD interaction in Omicron variants EG.5.1 and HV.1 were examined. The results demonstrated that 25 and 50 μg/mL of theaflavin led to a higher number of 293T-ACE2 cells with reduced RBD binding for the Omicron EG.5.1 variant (Figure 4A,B). In contrast, for the Omicron HV.1 variant, only the 50 μg/mL concentration of theaflavin had a similar effect (Figure 4C,D). These findings imply that at 50 μg/mL, theaflavin may disrupt the virus’s attachment process, potentially preventing infection initiation.

### 2.5. Effect of Theaflavin 3-Gallate on the ACE2–RBD Interaction of Omicron EG.5.1 and HV.1 Variants

Previous research has demonstrated that theaflavin 3-gallate can inhibit the binding between ACE2 and wild-type RBD spike protein [21]. This investigation was extended to assess its ability to inhibit ACE2–RBD interaction in Omicron EG.5.1 and HV.1 variants using dual-color flow cytometry. Results showed that 25 and 50 μg/mL of theaflavin 3-gallate increased the number of 293T-ACE2 cells with low RBD binding for both the Omicron EG.5.1 (Figure 5A,B) and HV.1 (Figure 5C,D) variants, indicating effective inhibition of ACE2–RBD binding. Flow cytometry analysis demonstrated that theaflavin 3-gallate exerted a potent inhibitory effect, resulting in a marked reduction in ACE2–RBD interaction across all tested variants and mutations.

### 2.6. Interaction of Theaflavin or Theaflavin 3-Gallate with RBD of Omicron EG.5.1 and HV.1 Variants via Molecular Docking

The notable increase in 293T-ACE2 cells with reduced RBD binding prompted an analysis of theaflavin’s interaction with the RBD of Omicron variants EG.5.1 and HV.1 through molecular docking. Theaflavin (ZINC number: ZINC3978446) exhibited lower idock scores in the RBD binding area for the SARS-CoV-2 Omicron EG.5.1 (−7 kcal/mol) and HV.1 (−7 kcal/mol) variants. Similarly, theaflavin 3-gallate (ZINC number: ZINC3978504) showed even lower idock scores in the RBD binding area for both Omicron EG.5.1 (−8 kcal/mol) and HV.1 (−8 kcal/mol) variants (Figure 6A). Detailed binding modes for theaflavin on Omicron EG.5.1 (Figure 6B) and HV.1 (Figure 6C) variants, as well as theaflavin 3-gallate on Omicron EG.5.1 (Figure 6D) and HV.1 (Figure 6E) variants, were identified. Hydrophobic interactions and additional hydrogen bonds played significant roles in binding. Molecular docking studies indicated that both theaflavin and theaflavin 3-gallate form stable interactions with key residues in the RBD, which are crucial for ACE2 binding. These interactions likely enhance the compounds’ capacity to inhibit viral entry by obstructing the ACE2–RBD interface.

## 3. Discussion

The evolution of SARS-CoV-2 is exemplified by the Omicron variants BA.1, EG.5.1, and HV.1. BA.1, first detected in November 2021, marked a significant shift in the pandemic, which was characterized by high transmissibility and immune evasion. As the virus continued to evolve, EG.5.1 (nicknamed “Eris”) emerged in 2023, which was followed closely by HV.1. These newer variants share many mutations with BA.1 but have acquired additional changes, particularly in the spike protein’s RBD. EG.5.1 and HV.1 are closely related, with HV.1 appearing to be a direct descendant of EG.5.1, featuring two additional mutations (F157L and L452R). Notably, the L452R mutation in HV.1 has been associated with increased transmissibility in previous variants. While BA.1 has some unique RBD mutations (Q493R, G496S), EG.5.1 and HV.1 share several new RBD mutations (V445P, F456L, N460K, F486P, F490S) that may enhance their ability to evade immunity. These evolutionary changes highlight the virus’s ongoing adaptation, potentially impacting transmission dynamics and immune escape capabilities [20,23,24]. As these variants continue to circulate and evolve, ongoing monitoring and research are crucial to understand their impact on public health and guide appropriate responses. GB-2, derived from Tian Shang Sheng Mu at Taiwan’s Chiayi Puzi Peitian Temple, is a traditional Chinese medicine formulation commonly used against SARS-CoV-2 [15,17,21]. Studies show it impedes Omicron BA.1 pseudovirus entry and disrupts ACE2 binding to RBD featuring specific mutations (N501Y, K417N, E484A, G339D, Q493R, G496S, Q498R) [15]. Our research further reveals GB-2’s capacity to interfere with ACE2–Spike protein interactions in newer variants EG.5.1 and HV.1. These findings indicate GB-2’s potential as a broad-spectrum inhibitor of various Omicron strains.

Hanai’s earlier research utilized computational analysis to examine the binding interactions between the Omicron spike protein’s receptor-binding domain and the ACE-2 receptor. The findings revealed that the Omicron variant exhibits higher binding energy compared to other variants, implying greater transmissibility. Although initial therapeutic compounds failed to effectively disrupt this interaction, altered catechins, such as Gallo catechin gallate, demonstrated potential in preventing the virus from binding to ACE-2 receptors [25]. Similarly, Alanzi et al. conducted virtual screenings to identify natural inhibitors for SARS-CoV-2 variants. They found that catechin, along with other flavonoids, showed strong binding affinities to the spike proteins of both Delta and Omicron variants, forming stable complexes and indicating its potential as an antiviral agent [26]. However, these findings are based on computational predictions and lack experimental validation. Our research further indicated that 25 and 50 μg/mL of (+)-catechin did not significantly affect the number of 293T-ACE2 cells binding to the RBD of the EG.5.1 variant. While computational studies suggest that catechins, especially modified forms like Gallo catechin gallate, could inhibit the binding of Omicron variants to ACE-2 receptors, experimental results showed no significant effect on RBD binding at the tested concentrations. This discrepancy underscores the need for more experimental research to validate the practical antiviral efficacy of catechins against Omicron variants.

Previous research indicates theaflavin may possess antiviral properties against SARS-CoV-2 and its Omicron variants. Computational analysis by Patel et al. suggests it could inhibit the main protease (Mpro) of both original and Omicron strains [27]. Goc et al. found a mixture containing theaflavin inhibited the RNA-dependent RNA polymerase (RdRp) complex in these viruses [28]. Our experiments showed theaflavin reduced RBD binding to 293T-ACE2 cells for Omicron EG.5.1 at 25 and 50 μg/mL concentrations, while only the higher dose was effective for Omicron HV.1. Molecular docking simulations yielded favorable scores (−7 kcal/mol) for theaflavin in the RBD binding area of both variants. These findings suggest theaflavin’s potential as an anti-SARS-CoV-2 agent across different variants. However, researchers emphasize the need for in vivo experiments and clinical trials to validate its practical effectiveness in COVID-19 prevention or treatment.

T3G (theaflavin 3-gallate), identified in black tea, is gaining attention as a potent adversary in the battle against SARS-CoV-2 and its evolving forms. Recent investigations highlight its potential to suppress various viral elements effectively. Studies illustrate T3G’s strong attachment to the SARS-CoV-2 spike protein RBD, possessing an impressively minimal KD of 1.3 nM, hinting at its capability to disrupt ACE2 binding, thus possibly thwarting viral entrance [29]. T3G also blocks SARS-CoV-2’s main protease (Mpro) with an IC50 of 18.48 μM and has shown to decrease viral loads by 75% in lab conditions [30]. Computational docking analyses suggest that T3G might attach more robustly to the RNA-dependent RNA polymerase (RdRp) than certain existing drugs [31]. It is noteworthy that T3G inhibits interactions between ACE2 and mutated RBDs found in Beta and Epsilon strains [21]. Simulations propose T3G’s ability to concurrently impede multiple SARS-CoV-2 targets [32]. Moreover, T3G curbs the protein expression of ACE2 and TMPRSS2 in test tubes and animal models without notable toxicity [33]. Data indicate that 25 and 50 μg/mL of T3G reduce RBD binding in 293T-ACE2 cells for the Omicron EG.5.1 and HV.1 subvariants, showcasing effective ACE2–RBD binding inhibition (Figure 5A,B). Additionally, T3G achieves lower idock scores in the RBD binding regions for these variants (−8 kcal/mol) (Figure 6A), underscoring its substantial inhibitory capability against SARS-CoV-2 and certain subvariants, including recent Omicron variants, by targeting a spectrum of viral and cellular targets. The specific inhibition of ACE2–RBD binding in Omicron EG.5.1 and HV.1 variants is particularly significant. Nonetheless, while these findings are promising, they primarily derive from computational or in vitro studies and lack clinical validation. The effect of T3G on newer variants demands more detailed exploration. Additionally, its bioavailability and pharmacokinetics in humans require further assessment. To confirm T3G’s effectiveness as a therapeutic against SARS-CoV-2 and its variants, more expansive in vivo research and clinical trials are essential. Future studies should aim to overcome the current limitations to fully ascertain T3G’s effectiveness, safety, and practical usage in managing COVID-19 and its ongoing mutations.

## 4. Materials and Methods

### 4.1. Preparation of GB-2

GB-2, which originated from the Tian Shang Sheng Mu of Chiayi Puzi Peitian Temple in Taiwan (the Tian Shang Sheng Mu of Chiayi Puzi Peitian Temple in Taiwan), was prepared following established protocols through the Chinese pharmacy department of the Chiayi Chang Gung Memorial Hospital [15]. In brief, the study utilized 10 g of *Glycyrrhiza uralensis* Fisch. sourced from Hangjinqi, Inner Mongolia (specimen No.7H-E014, from Hangjinqi, Inner Mongolia, China) and 25 g of *Camellia sinensis* var. *assamica* (J.W.Mast.) Kitam. from Chiayi City, Taiwan (specimen No.7H-Y059, provided by Chang Gung Memorial Hospital, from Chiayi city, Taiwan). Dr. Yu-Shih Lin, a pharmacist at Chiayi Chang Gung Memorial Hospital’s Chinese pharmacy department, authenticated the herbal samples. Preparation involved combining these herbs in 2000 mL of water, simmering for 25 min in a thermal container, then filtering out solids using filter paper. This method produced 1500 mL of herbal extract. For laboratory research, the extract underwent further processing. It was concentrated under vacuum to create a 6-gram viscous substance, which was then stored at −80 °C. This concentrated form was subsequently diluted as required for various experimental assays to achieve the desired concentrations.

### 4.2. Quality Control of GB-2

Quality control of GB-2 was conducted using an Agilent 1100 High-performance Liquid Chromatography (HPLC) system. The fingerprint chromatography for the approved formulation employed an HPLC method with a C18 column (Discovery^®^, 4.6 mm × 150 mm, 5 μM). Mobile phase A consisted of 0.3% phosphoric acid, while B was 100% acetonitrile with a gradient program of 90:10 at 0 min, 80:20 at 16.2 min, 40:60 at 23.4 min, and returning to 90:10 at 23.76 min. Operating conditions included a flow rate of 0.7 mL/min, detection wavelength of 278 nm, and column temperature of 30 °C. The index compounds used were theaflavin 3-gallate (sourced from ChromaDex, Irvine, CA, USA, Lot Number: 000258-841), theaflavin (from ChromaDex, Irvine, CA, USA, Lot Number: 00020252-201), and (+)-catechin (from Sigma-Aldrich, St. Louis, MO, USA, Batch Number: BCCC3128). For glycyrrhizic acid (sourced from ChromaDex, Irvine, CA, USA) detection, a C18 column was used with a mobile phase of 36% acetonitrile and 64% of 2% acetic acid solution, flow rate of 0.6 mL/min, wavelength of 278 nm, and column temperature of 25 °C.

### 4.3. Cell Culture and Treatment

The 293T cells (human embryonic kidney cell line) were sourced from the Bioresource Collection and Research Center in Taiwan. These cells were grown in Dulbecco’s Modified Eagle’s medium (DMEM, Invitrogen Corp., Carlsbad, CA, USA, Cat. Number: 11965-048), enriched with 10% FBS, maintained at 37 °C and 5% CO_2_. The pCEP4-myc-ACE2 plasmid was generously provided by Erik Procko (Addgene plasmid #141185; http://n2t.net/addgene:141185; RRID:Addgene_141185, accessed on 7 December 2020). Another plasmid, pcDNA3-SARS-CoV-2-S-RBD-sfGFP, was also provided by Erik Procko (Addgene plasmid #141184; http://n2t.net/addgene:141184; RRID:Addgene_141184, accessed on 24 July 2020). The pLENTI_hACE2_PURO plasmid came from Raffaele De Francesco (Addgene plasmid #155295; http://n2t.net/addgene:155295; RRID:Addgene_155295, accessed on 23 February 2021). Didier Trono supplied the psPAX2 plasmid (Addgene plasmid #12260; http://n2t.net/addgene:12260; RRID:Addgene_12260, accessed on 23 February 2021). The pLV-eGFP plasmid was provided by Pantelis Tsoulfas (Addgene plasmid #36083; http://n2t.net/addgene:36083; RRID:Addgene_36083, accessed on 17 February 2021). Before treatment, 293T cells were cultured until they reached 60–70% confluence. The existing medium was replaced with fresh medium containing the specified compounds dissolved in water at the indicated concentrations. Cells treated with only water served as the control group, while untreated 293T cells were used as blank controls.

### 4.4. Site-Directed Mutagenesis

Amino acid changes (G339H, R346T, L368I, S371F, S373P, S375F, T376A, K417N, N440K, V445P, G446S, F456L, N460K, S477N, T478K, E484A, F486P, F490S, Q498R, N501Y, Y505H in the EG.5.1 variant; G339H, R346T, L368I, S371F, S373P, S375F, T376A, K417N, N440K, V445P, G446S, L452R, F456L, N460K, S477N, T478K, E484A, F486P, F490S, Q498R, N501Y, Y505H in the HV.1 variant) were generated in the pcDNA3-SARS-CoV-2-S-RBD-sfGFP plasmid utilizing the QuikChange Lightning Site-Directed Mutagenesis kit (Agilent Technologies Inc., Santa Clara, CA, USA, Cat. Number: 210519). This was achieved by strictly adhering to the manufacturer’s protocol.

### 4.5. Flow Cytometry Analysis of ACE2–Spike Protein Binding

ACE2-S binding analysis via flow cytometry followed established methods [15,21,22,34]. In summary, 293T cells were transfected with pCEP4-MYC-ACE2 or pcDNA3-SARS-CoV-2-S-RBD-sfGFP plasmids (EG.5.1 or HV.1 variants) using Lipofectamine 2000 (ThermoFisher, Cat. Number: 11668-019, Waltham, MA, USA). The DNA concentration was 500 ng per mL of culture, with 2 × 106 cells/mL. After 48 h, the RBD-sfGFP-containing medium was harvested from cells transfected with the pcDNA3-SARS-CoV-2-S-RBD-sfGFP plasmids. Cells transfected with pCEP4-MYC-ACE2 were treated with drugs for 24 h and, then washed with cold PBS-BSA. These cells were incubated on ice for 30 min with the RBD-sfGFP medium and anti-MYC Alexa 647 antibody (clone 9B11, Cell Signaling Technology, Danvers, MA, USA). After washing, the cells were analyzed using a BD FACSCanto flow cytometer (BD, Franklin Lakes, NJ, USA). The analysis focused on cell populations expressing MYC-tagged ACE2 or GFP-tagged RBD following compound treatment. Using fluorescence-activated cell sorting, either high or low binding to different RBD variants was observed. To analyze the results, the ratio of treated cell populations (both low and high groups) to their corresponding vehicle-treated controls (blank group) was calculated. This ratio represents the relative cell numbers in each group, normalized against the blank group, allowing assessment of the impact of different compound concentrations on RBD binding.

### 4.6. Structure Preparation

The 3D structures of the SARS-CoV-2 spike protein (NCBI Reference Sequence: YP_009724390.1) for various Omicron variants, incorporating mutations (G339H, R346T, L368I, S371F, S373P, S375F, T376A, K417N, N440K, V445P, G446S, F456L, N460K, S477N, T478K, E484A, F486P, F490S, Q498R, N501Y, Y505H in the EG.5.1 variant; G339H, R346T, L368I, S371F, S373P, S375F, T376A, K417N, N440K, V445P, G446S, L452R, F456L, N460K, S477N, T478K, E484A, F486P, F490S, Q498R, N501Y, Y505H in the HV.1 variant) were generated via homology modeling. This modeling utilized Modeller [20] in combination with UCSF Chimera [21] and SWISS-MODEL [22].

### 4.7. Molecular Docking and Virtual Screening

For molecular docking and virtual screening, the idock method was employed. Molecular docking and virtual screening were carried out using idock, which was downloaded from GitHub (Idock 2.0) [35] and executed on a local Linux machine. Nine docking poses were produced for each variant, and the top-scoring poses were ranked by idock.

### 4.8. Statistical Analyses

All reported values represent the means ± standard error of mean (SEM) from triplicate samples. Each experiment was conducted at least three times. For testing the normality, we apply the Shapiro–Wilk test with all data values. The null hypothesis is that data are distributed normally and the *p*-value = 0.059 did not show a significant departure from normality. Comparisons between two groups were assessed using an unpaired two-tailed Student’s *t*-test. The Tukey test was utilized to determine the significance of pairwise comparisons among multiple groups. A *p*-value less than 0.01 was considered statistically significant for all analyses. Statistical computations were performed using SPSS version 13.0 (SPSS Inc., Chicago, IL, USA).

## 5. Conclusions

In conclusion, the study confirms the inhibitory effects of GB-2, theaflavin, and theaflavin 3-gallate on the ACE2–RBD interaction across different Omicron variants. The high-performance liquid chromatography analysis validated the presence of these compounds in GB-2, with theaflavin 3-gallate showing particularly potent effects in reducing ACE2–RBD binding, as evidenced by flow cytometry and molecular docking results. Theaflavin and theaflavin 3-gallate notably decreased the binding of the RBD to ACE2 in the Omicron EG.5.1 and HV.1 variants, suggesting their potential as therapeutic agents in preventing COVID-19 transmission and infection. These findings underline the need for further research to explore the clinical relevance and therapeutic potential of these compounds against SARS-CoV-2 and its variants.

## Figures and Tables

**Figure 1 ijms-25-09498-f001:**
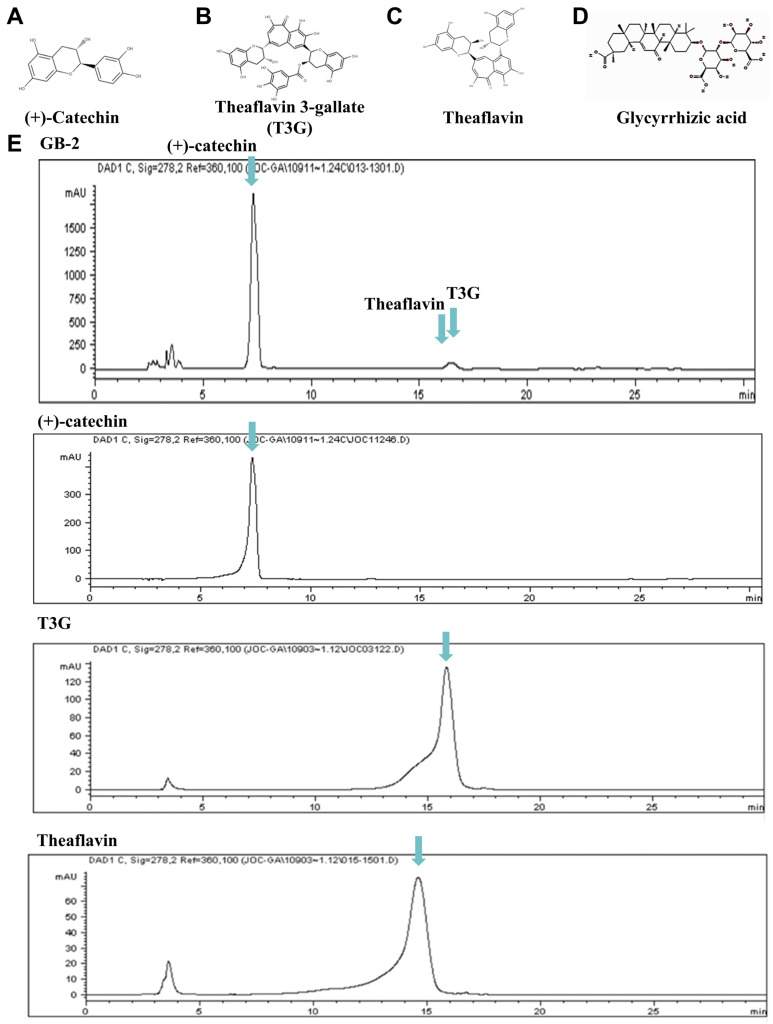
HPLC analysis of GB-2 and its components. The structures of the following compounds are shown: (**A**) (+)-catechin, (**B**) theaflavin 3-gallate (T3G), (**C**) theaflavin, and (**D**) glycyrrhizic acid. (**E**) displays the HPLC fingerprint of GB-2, with (+)-catechin, T3G, and theaflavin used as reference compounds. (**F**) presents the HPLC chromatogram of GB-2 and glycyrrhizic acid.

**Figure 2 ijms-25-09498-f002:**
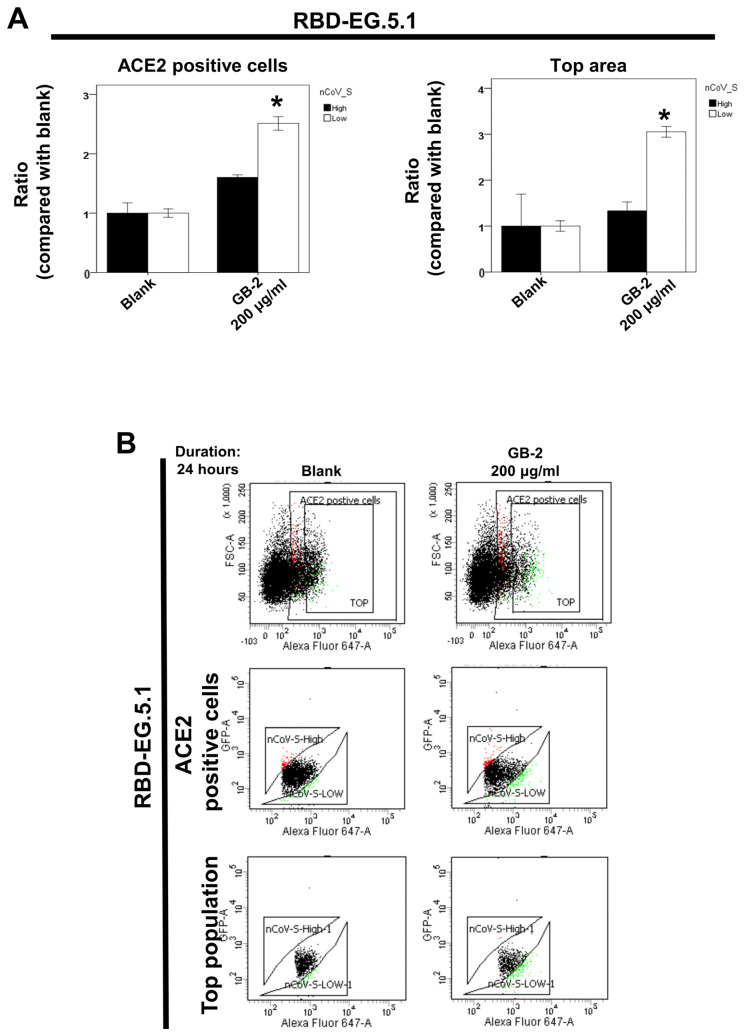
Effect of GB-2 on the interaction between ACE2 and the SARS-CoV-2 spike protein in different Omicron variants (EG.5.1 and HV.1). Flow cytometry analysis of ACE2–Spike protein binding was performed. The 293T cells with pCEP4-MYC-ACE2 plasmid were incubated with RBD-sfGFP from different Omicron variants and co-stained with anti-MYC Alexa 647 to detect surface ACE2. The ACE2-positive population was divided into top (nCoV-S-High sort) and bottom (nCoV-S-Low sort) subsets based on RBD-sfGFP fluorescence (Black dots: population without nCoV-S-High and nCoV-S-Low; Red dots: population with nCoV-S-High; Green dots: population with nCoV-S-Low). Results for the top and bottom populations in Omicron EG.5.1 (**A**) and HV.1 (**C**) variants are presented. (**B**,**D**) Quantitative results of nCoV-S-High sort and nCoV-S-low sort, which were presented as ratio compared with blank, in the top population or ACE2-positive population. Error bars represent mean ± S.E.M. Asterisks (*) indi-cate samples significantly different from the control group with *p* < 0.01, n = 3. Results are repre-sentative of at least three independent experiments.

**Figure 3 ijms-25-09498-f003:**
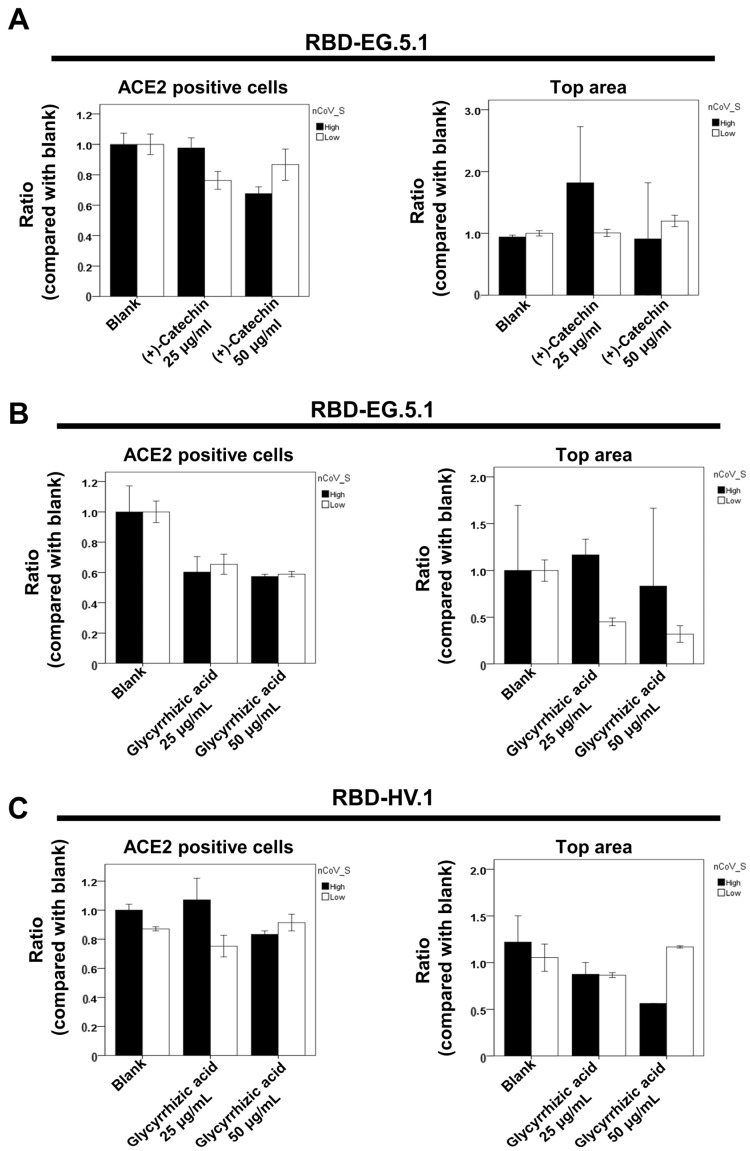
Effect of (+)-catechin and glycyrrhizic acid on the interaction between ACE2 and the SARS-CoV-2 spike protein in different Omicron variants. Flow cytometry analysis of ACE2–Spike protein binding was performed. The 293T cells with pCEP4-MYC-ACE2 plasmid were incubated with RBD-sfGFP from Omicron variants and co-stained with anti-MYC Alexa 647 to detect surface ACE2. The ACE2-positive population was divided into top (nCoV-S-High sort) and bottom (nCoV-S-Low sort) subsets based on RBD-sfGFP fluorescence. Results for the top and bottom populations in Omicron EG.5.1 (**A**,**B**) and HV.1 (**C**) variants are presented. Error bars represent mean ± S.E.M. Results are representative of at least three independent experiments.

**Figure 4 ijms-25-09498-f004:**
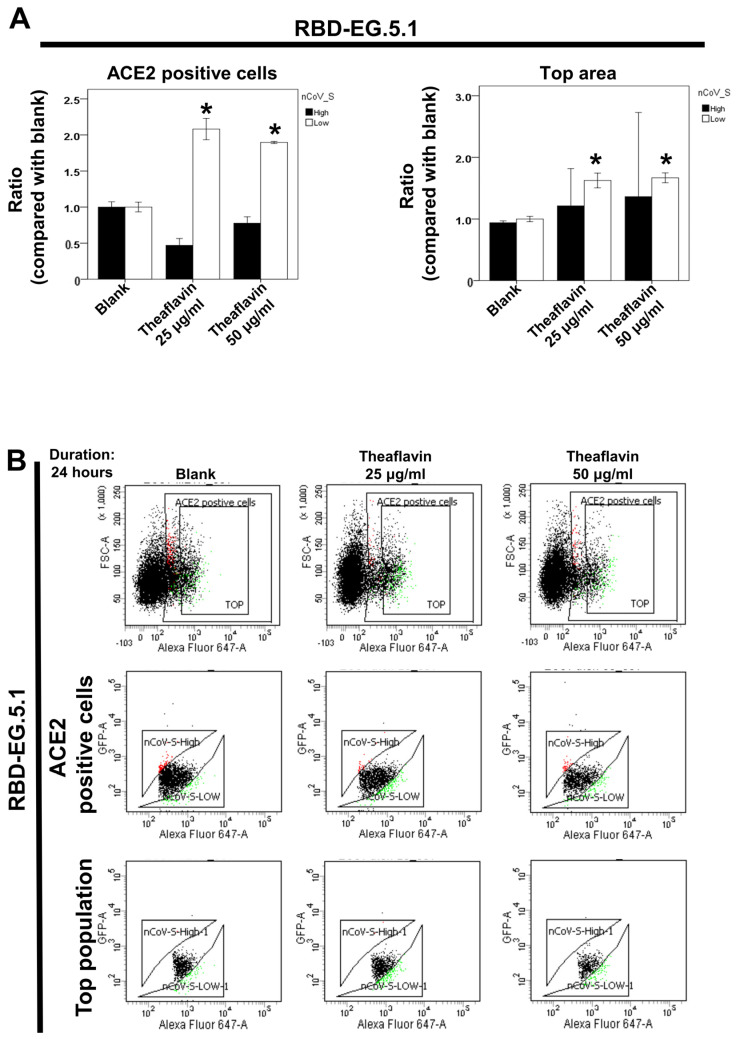
Effect of theaflavin on the interaction between ACE2 and the SARS-CoV-2 spike protein in different Omicron variants (EG.5.1 and HV.1). Flow cytometry analysis of ACE2–Spike protein binding was performed. The 293T cells with pCEP4-MYC-ACE2 plasmid were incubated with RBD-sfGFP from different Omicron variants and co-stained with anti-MYC Alexa 647 to detect surface ACE2. The ACE2-positive population was divided into top (nCoV-S-High sort) and bottom (nCoV-S-Low sort) subsets based on RBD-sfGFP fluorescence (Black dots: population without nCoV-S-High and nCoV-S-Low; Red dots: population with nCoV-S-High; Green dots: population with nCoV-S-Low). Results for the top and bottom pop-ulations in Omicron EG.5.1 (**A**) and HV.1 (**C**) variants are presented. (**B**,**D**) Quantitative results of nCoV-S-High sort and nCoV-S-low sort, which were presented as ratio compared with blank, in the top population or ACE2-positive population. Error bars represent mean ± S.E.M. Asterisks (*) indi-cate samples significantly different from the control group with *p* < 0.01, n = 3. Results are repre-sentative of at least three independent experiments.

**Figure 5 ijms-25-09498-f005:**
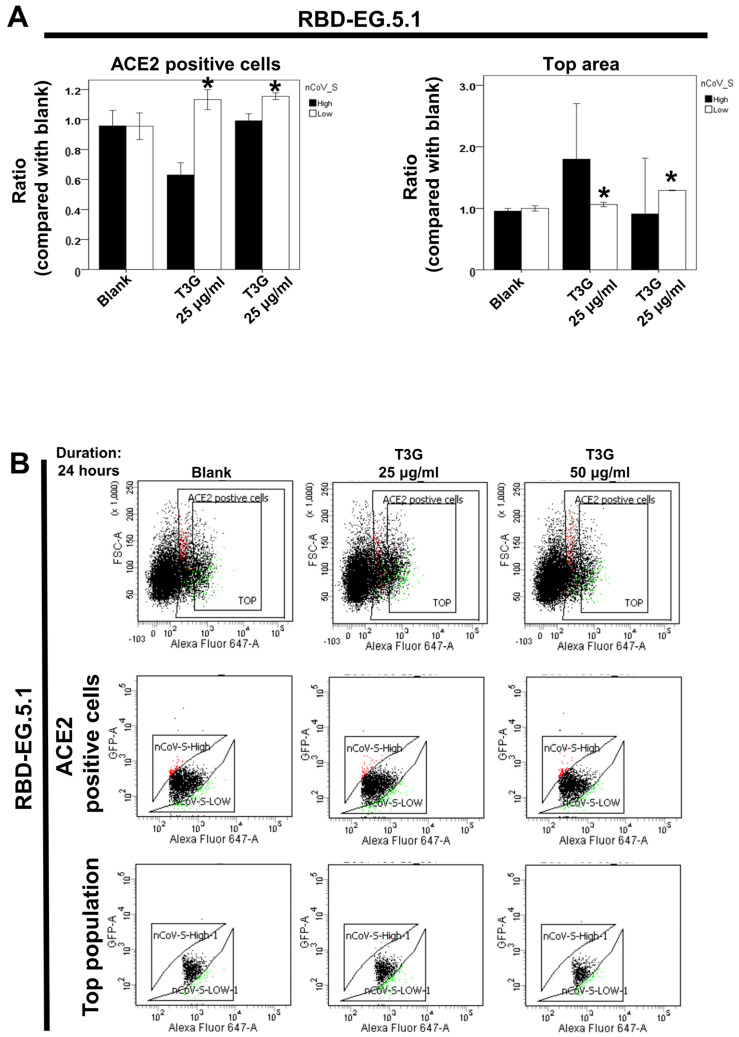
Effect of theaflavin 3-gallate on the interaction between ACE2 and the SARS-CoV-2 spike protein in different Omicron variants (EG.5.1 and HV.1). Flow cytometry analysis of ACE2–Spike protein binding was performed. The 293T cells with pCEP4-MYC-ACE2 plasmid were incubated with RBD-sfGFP from different Omicron variants and co-stained with anti-MYC Alexa 647 to detect surface ACE2. The ACE2-positive population was divided into top (nCoV-S-High sort) and bottom (nCoV-S-Low sort) subsets based on RBD-sfGFP fluorescence(Black dots: population without nCoV-S-High and nCoV-S-Low; Red dots: population with nCoV-S-High; Green dots: population with nCoV-S-Low). Results for the top and bottom pop-ulations in Omicron EG.5.1 (**A**) and HV.1 (**C**) variants are presented. (**B**,**D**) Quantitative results of nCoV-S-High sort and nCoV-S-low sort, which were presented as ratio compared with blank, in the top population or ACE2-positive population. Error bars represent mean ± S.E.M. Asterisks (*) indi-cate samples significantly different from the control group with *p* < 0.01, n = 3. Results are repre-sentative of at least three independent experiments.

**Figure 6 ijms-25-09498-f006:**
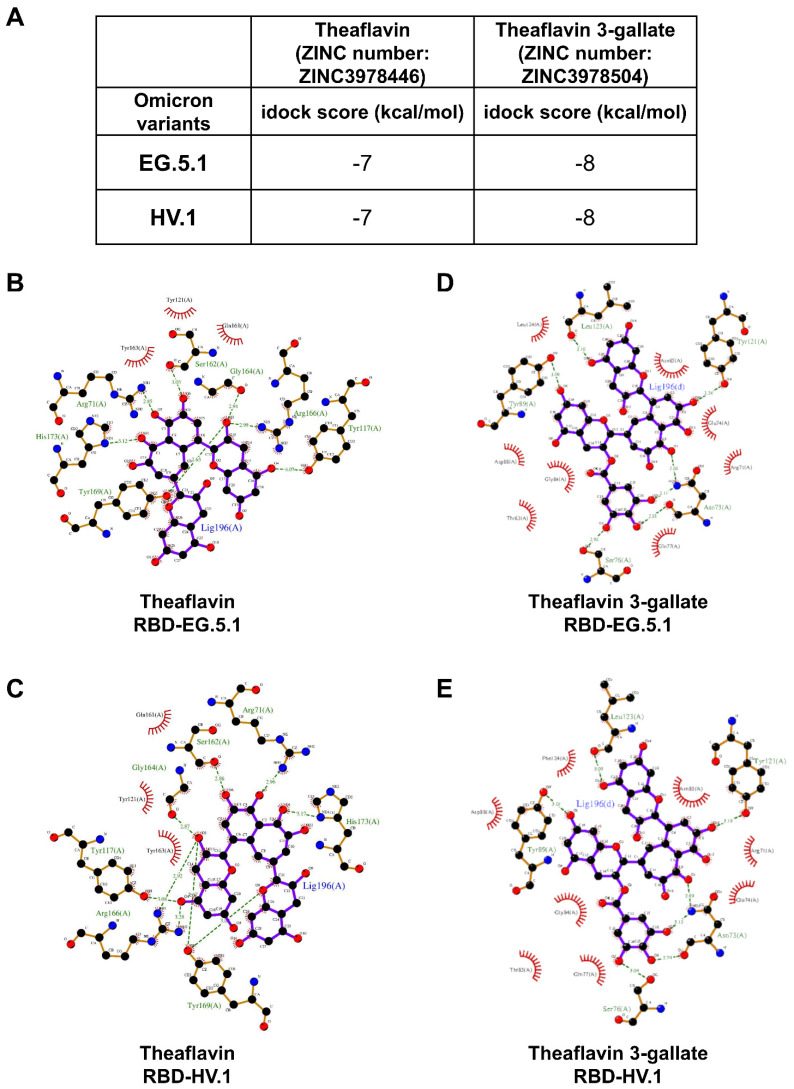
Interaction of theaflavin and theaflavin 3-gallate with the RBD of different Omicron variants through molecular docking. (**A**) Table of idock scores. (**B**–**E**) The 2D interaction diagrams illustrate the contact models between theaflavin and the RBD of EG.5.1 (**B**) and HV.1 (**C**) variants, which were created using the idock method. Similarly, diagrams for theaflavin 3-gallate interacting with the RBD of EG.5.1 (**D**) and HV.1 (**E**) variants are also generated through the idock method. The diagrams analyze and display the relative distances between amino acid residues and theaflavin or theaflavin 3-gallate, which were visualized using LigPlot+. Atoms of carbon, oxygen, nitrogen, and fluorine are represented as black, red, blue, and green circles, respectively. The covalent bonds linking theaflavin or theaflavin 3-gallate to amino acid residues in the RBD are colored in purple and orange. Light green dotted lines mark the hydrogen bond distances (in Å) between the compounds’ functional groups and the amino acid residues. Hydrophobic interactions are indicated with the names of the involved amino acid residues, which are highlighted by dark red markers pointing toward the corresponding functional group of theaflavin or theaflavin 3-gallate.

## Data Availability

All data generated or analyzed during this study are indicated in this article (with no patient data). The datasets generated during and/or analyzed during the current study are available from the corresponding author on reasonable request.

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
