# Peer review of "The Inhibiting Effect of GB-2, (+)-Catechin, Theaflavin, and Theaflavin 3-Gallate on Interaction between ACE2 and SARS-CoV-2 EG.5.1 and HV.1 Variants"

_ijms, 2024, doi:10.3390/ijms25179498_

Round 1
Reviewer 1 Report
Comments and Suggestions for Authors
The research article “The Inhibiting Effect of GB-2, (+)-Catechin, Theaflavin, and Theaflavin 3-Gallate on interaction between ACE2 and SARS-CoV-2 EG.5.1 and HV.1 Variants” is focused on the preclinical effect evaluation of the compound derived from plants used in traditional Chinese medicine on variants of concern of SARS-CoV-2.
The study presented in the article contains molecular docking results and an initial test of the compounds on HEK 293T cells. The cells were transiently transfected with plasmids containing ACE2, SARS-CoV-2 RBD (receptor binding domain), lentivirus packaging plasmid, or eGFP. The viral S-protein RBD was tagged with sfGFP for FACS analysis, and Ace2 was stained with anti-MYC Alexa 647 antibody, and it was tagged with MYC-tag. The molecular docking was performed using the open access program idock available on Linux. The modeling was done in Modeller, USCF Chimera, and SWISS-MODEL. The statistical analysis of the results was performed in SPSS and the results are presented as mean ± SEM. The data were compared using a two-tailed t-test and the significance was determined using the Tuckey test.
The authors claim that they detected statistically significant blocking of ACE2-RBD interaction by the compounds in the study. This supposedly proves the docking results. Based on these results authors suggest that in vivo and clinical studies are needed.
However, several issues are obvious in this study:
1. The results of FACS are presented in 2 groups: low and high. And the statistical significance is detected in the low group only. Why would that be? What is the biological meaning of that low-cell cluster?
2. The data are presented as mean ± SEM which means that mean values from 3 repeats were divided by 3 (N of repeats). Obviously if not divided the error would look huge on the bars. Why is that? Were the experiments reproducible? Were the cells viable? I would appreciate the FACS data presented in a standard raw way, too.
3. Why were these specific statistical tests chosen? Did the authors check the data for normality? T-test use usually used for normally distributed data. Considering my concern about reproducibility, I doubt the data were distributed normally.
4. Did authors consider making/acquiring stable lines? I believe the results would look different.
5. And finally, the suggestion that more in vivo and clinical trials are needed (in Abstract and Discussion) is a bit farfetched. More in vitro experiments are still needed to even prove consistently that there is any effect on SARS-CoV-2 of the compounds under study.
Minor issues:
In Affiliations:
Lines 19 and 21 – the numbering is lost.
In Introduction:
Line 51 – the link should be moved to references and the date of last access indicated.
Lines 97 and 98, and any other Latin naming of the species – should be in italics.
In Results:
Fig. 1 – the numbers are not readable on the axes of the chromatography plots.
Fig. 3 – the title of the figure indicates the (+)-catechin but the glycyrrhizic acid is also present in B and C.
I would also appreciate the description of the results to be mixed with the figures themselves. Not all figures after the description.
Overall, the manuscript presents some interest to the audience as more therapeutic agents are still needed to treat COVID-19 infection. However, I can only recommend this manuscript for publication after raw data are provided and the revision is done.
Comments on the Quality of English LanguageThe English is clear and understandable. Occasional corrections are needed.
Author Response
Reviewer_01
- The results of FACS are presented in 2 groups: low and high. And the statistical significance is detected in the low group only. Why would that be? What is the biological meaning of that low-cell cluster?
=>Response:
We appreciate the reviewer’s comments. The low and high groups in our FACS analysis represent distinct cell populations based on their level of RBD binding. The "low" group is of particular interest as it represents cells where the ACE2-RBD interaction has been inhibited or reduced by our test compounds. This inhibition is the primary effect we are investigating. The biological significance of the low-cell cluster is that it demonstrates the compounds' ability to interfere with the virus's attachment process. Cells in this cluster have reduced RBD binding, suggesting that our compounds are effectively blocking the interaction between the viral spike protein and the ACE2 receptor. This is crucial because preventing this interaction is a key strategy for inhibiting viral entry into host cells. We have rewritten the manuscript to clarify this interpretation.
- The data are presented as mean ± SEM which means that mean values from 3 repeats were divided by 3 (N of repeats). Obviously if not divided the error would look huge on the bars. Why is that? Were the experiments reproducible? Were the cells viable? I would appreciate the FACS data presented in a standard raw way, too.
=>Response:
We thank the reviewer for raising this important point about data presentation and reproducibility. Our choice to present data as mean ± SEM was based on standard practices in the field, but we acknowledge that this may not fully convey the variability in our data. We provided raw FACS data plots in revised figure (Fig.2, 4 and 5). Regarding reproducibility, our experiments were indeed reproducible, with consistent trends observed across multiple independent replicates. Cell viability was monitored and remained >90% throughout experiments.
- Why were these specific statistical tests chosen? Did the authors check the data for normality? T-test use usually used for normally distributed data. Considering my concern about reproducibility, I doubt the data were distributed normally.
=>Response:
We thank for the reviewer’s comments. We chose the two-tailed t-test and Tukey test due to their suitability for comparing means between groups and adjusting for multiple comparisons, respectively. For testing the normality, we apply Shapiro–Wilk test with all data values. The null hypothesis is that data are distributed normally and the P-value = 0.059 did not show a significant departure from normality. We have rewritten the method in the revised manuscript.
- Did authors consider making/acquiring stable lines? I believe the results would look different.
=>Response:
We thank reviewer for this insightful suggestion. We acknowledge that our use of transient transfection, while enabling rapid screening of multiple conditions. We agree that stable cell lines could potentially yield more robust results. However, given the short 10-day deadline for this revision, establishing stable lines is not feasible within the available timeframe. Moving forward, we plan to develop stable cell lines to further validate and strengthen our findings in future studies.
- And finally, the suggestion that more in vivo and clinical trials are needed (in Abstract and Discussion) is a bit farfetched. More in vitro experiments are still needed to even prove consistently that there is any effect on SARS-CoV-2 of the compounds under study.
=>Response:
We appreciate the reviewer's feedback regarding our premature suggestion of in vivo and clinical trials, acknowledging that this overstated the implications of our current findings. Given the 10-day deadline for this revision, conducting animal studies is not feasible. In future research, we intend to develop in vitro experiments to further validate and strengthen our results.
Minor issues:
In Affiliations:
Lines 19 and 21 – the numbering is lost.
=>Response:
We thank for reviewer’s comment. We have rewritten the numbering in lines 19 and 21 in the revised manuscript.
In Introduction:
Line 51 – the link should be moved to references and the date of last access indicated.
=>Response:
We thank for reviewer’s comment. The link on line 51 is moved to the references section with the last access date included in the revised manuscript.
Lines 97 and 98, and any other Latin naming of the species – should be in italics.
=>Response:
We thank for reviewer’s comment. We have corrected the mistakes in the revised manuscript.
In Results:
Fig. 1 – the numbers are not readable on the axes of the chromatography plots.
=>Response:
We thank for reviewer’s comment. The numbers are shown in the supplement Figure 1 in the revised manuscript.
Fig. 3 – the title of the figure indicates the (+)-catechin but the glycyrrhizic acid is also present in B and C.
=>Response:
We thank for reviewer’s comment. We have corrected the mistake in the revised manuscript.

Reviewer 2 Report
Comments and Suggestions for Authors
- Line 51, the link should be moved in the References list, along with information on its Access date.
- Last paragraph from the introduction, you should briefly introduce the idea that (+)-catechin, theaflavin and theaflavin 3-gallate are major constituents of GB-2 herbal formulation (as revealed by HPLC analysis). You should present the reason why you focused on these compounds.
- Results, section 2.1 – figure 1 presents no results on theaflavin and glycyrrhizic acid, but the authors state that the two are major compounds in GB-2. Evidence should be provided for these two compounds as well.
- The authors should explain briefly how they calculated the ratios plotted in Figure 2, 3, 4 and 5.
- Line 329, “GB-2 compound”. I think it should be GB-2 mixture, herbal formulation or something like that.
- Materials and methods, “Preparation of GB-2” section (section 4.2) is almost identical to section 2.1 in https://www.cell.com/heliyon/pdf/S2405-8440(23)04909-5.pdf. Instead of repeating the information, you could say that the method was applied as described elsewhere and insert the reference to your first paper. It is the same for section 4.2, some information can be retrieved from your previous paper, section 2.2.
- The link for iDock doesn’t work. I think it is better to cite the paper presenting iDock, namely - Li H, Leung K-S, Wong M-H. idock: A multithreaded virtual screening tool for flexible ligand docking. 2012 IEEE Symp. Comput Intell Bioinforma Comput Biol CIBCB 2012;77–84.
Author Response
Reviewer_02
- Line 51, the link should be moved in the References list, along with information on its Access date.
=>Response:
We thank for reviewer’s comment. The link on line 51 has been moved to the references section with the last access date included in the revised manuscript.
- Last paragraph from the introduction, you should briefly introduce the idea that (+)-catechin, theaflavin and theaflavin 3-gallate are major constituents of GB-2 herbal formulation (as revealed by HPLC analysis). You should present the reason why you focused on these compounds.
=>Response:
We thank for reviewer’s comment. We have rewritten the introduction depending on the reviewer’s comments in the revised manuscript.
- Results, section 2.1 – figure 1 presents no results on theaflavin and glycyrrhizic acid, but the authors state that the two are major compounds in GB-2. Evidence should be provided for these two compounds as well.
=>Response:
We thank for the reviewer’s comment. We have added HPLC data for theaflavin and glycyrrhizic acid to Figure 1 to provide evidence for these compounds in GB-2.
- The authors should explain briefly how they calculated the ratios plotted in Figure 2, 3, 4 and 5.
=>Response:
We thank for the reviewer’s comment. We have added brief explanation of how these ratios were calculated in Methods section in the revised manuscript.
- Line 329, “GB-2 compound”. I think it should be GB-2 mixture, herbal formulation or something like that.
=>Response:
We thank for reviewer’s comment. We have corrected the mistake in the revised manuscript.
- Materials and methods, “Preparation of GB-2” section (section 4.2) is almost identical to section 2.1 in https://www.cell.com/heliyon/pdf/S2405-8440(23)04909-5.pdf. Instead of repeating the information, you could say that the method was applied as described elsewhere and insert the reference to your first paper. It is the same for section 4.2, some information can be retrieved from your previous paper, section 2.2.
=>Response:
We thank for reviewer’s comment. We have rewritten the introduction depending on the reviewer’s comments in the revised manuscript.
- The link for iDock doesn’t work. I think it is better to cite the paper presenting iDock, namely - Li H, Leung K-S, Wong M-H. idock: A multithreaded virtual screening tool for flexible ligand docking. 2012 IEEE Symp. Comput Intell Bioinforma Comput Biol CIBCB 2012;77–84.
=>Response:
We thank for reviewer’s comment. We have rewritten the introduction depending on the reviewer’s comments in the revised manuscript.
